# Early Increase in Blood–Brain Barrier Permeability in a Murine Model Exposed to Fifteen Days of Intermittent Hypoxia

**DOI:** 10.3390/ijms25053065

**Published:** 2024-03-06

**Authors:** Frederic Roche, Anne Briançon-Marjollet, Maurice Dematteis, Marie Baldazza, Brigitte Gonthier, Frederique Bertholon, Nathalie Perek, Jean-Louis Pépin

**Affiliations:** 1INSERM, SAINBIOSE U1059, Université Jean Monnet Saint-Étienne, Mines Saint Etienne, F-42023 Saint-Étienne, France; nathalie.perek@univ-st-etienne.fr; 2INSERM U1300, HP2 Laboratory, Université Grenoble Alpes, F-38042 Grenoble, France; anne.briancon-marjollet@univ-grenoble-alpes.fr (A.B.-M.); mdematteis@chu-grenoble.fr (M.D.); mariebaldazza@yahoo.fr (M.B.); brigitte.gonthier@univ-grenoble-alpes.fr (B.G.); jpepin@chu-grenoble.fr (J.-L.P.); 3Centre de Ressources Biologiques, CHU de Saint Etienne, F-42055 Saint-Étienne, France; frederique.bertholon@chu-st-etienne.fr

**Keywords:** blood–brain barrier, tight junction, aquaporins, intermittent hypoxia, mice model

## Abstract

Obstructive sleep apnea (OSA) is characterized by intermittent repeated episodes of hypoxia–reoxygenation. OSA is associated with cerebrovascular consequences. An enhanced blood–brain barrier (BBB) permeability has been proposed as a marker of those disorders. We studied in mice the effects of 1 day and 15 days intermittent hypoxia (IH) exposure on BBB function. We focused on the dorsal part of the hippocampus and attempted to identify the molecular mechanisms by combining in vivo BBB permeability (Evans blue tests) and mRNA expression of several junction proteins (zona occludens (ZO-1,2,3), VE-cadherin, claudins (1,5,12), cingulin) and of aquaporins (1,4,9) on hippocampal brain tissues. After 15 days of IH exposure we observed an increase in BBB permeability, associated with increased mRNA expressions of claudins 1 and 12, aquaporins 1 and 9. IH seemed to increase early for claudin-1 mRNA expression as it doubled with 1 day of exposure and returned near to its base level after 15 days. Claudin-1 overexpression may represent an immediate response to IH exposure. Then, after 15 days of exposure, an increase in functional BBB permeability was associated with enhanced expression of aquaporin. These BBB alterations are possibly associated with a vasogenic oedema that may affect brain functions and accelerate neurodegenerative processes.

## 1. Introduction

Obstructive sleep apnea (OSA) is characterized by repeated partial or complete upper airway obstruction during sleep. These obstructions alter alveolar ventilation, leading to episodic hypoxemia and hypercapnia as well as sleep fragmentation [1]. The high prevalence of this syndrome (4–20% depending on the population studied) raises the issue of public health screening, particularly in the elderly, the population group in which OSA prevalence is highest (25%). OSA is now recognized as a common clinical problem, and questions of prevalence, risk and prognosis are becoming increasingly important for diagnosis and treatment. The pathology has consequences, e.g., neurological impairments and cardiovascular disorders [2,3]. Over the last decades, a large number of studies have been conducted to explore potential mechanisms underlying the adverse effects of sleep apnea on the cardiovascular system and the central nervous system (CNS). Studies in adult patients with OSA showed the presence of changes in brain diffusivity in MRI studies along with the presence of increased intracranial pressure. Patients with sleep apnea showed structural and functional disorders of the CNS. Cognitive dysfunctions [4] comprise impairments in attention [5,6,7], executive function [8] and memory [9]. The link between OSA and brain structure alteration remains unclear despite dysfunction of the hippocampus, middle temporal gyrus and brainstem/cerebellum having been observed in patients [5]. Lim and Pack [10] proposed the hypothesis that a disruption of the blood–brain barrier (BBB), a barrier separating the blood from the brain, may contribute to cognitive impairments in OSA.

The BBB is a physical and metabolic barrier which separates the CNS from blood circulation. Endothelial cells are the main site of the BBB because they restrict and control the passage of molecules into and out of the brain. The endothelial cells of cerebral blood capillaries are interconnected by tight junctions and adherens junctions which form one of the main elements of the BBB [11]. TJs are composed of occluding, claudin and junction adhesion molecules. In addition, many other important cytoplasmic proteins, including ZOs, occludin and cingulin, contribute to maintain the BBB’s permeability [12]. Adherent junctions play a role in maintaining the structural stability of TJs and are essentially composed of vascular endothelial (VE)-cadherin [13]. In the BBB endothelium, it was suggested that claudin-3, claudin-5 and claudin-12 contribute to the tightness of this barrier [14]. Although rarely expressed at the normal blood–brain barrier (BBB), claudin-1 is expressed in pathological conditions [15]. Claudin-5 is the most enriched claudin in the vascular endothelium of the BBB. Its absence is associated with BBB integrity breakdown and it is also implied in the pathologies of stroke, traumatic brain injury and schizophrenia [13]. Claudin-3 was described to be involved in the induction and maintenance of the BBB [16,17]. Expression of claudin-12, an unusual member of the claudin family, has additionally been described at the BBB [18] and its expression in brain endothelial cells was recently confirmed [19]. Claudin-12 mediates the interaction of claudins with the cytoskeleton, by allowing the binding to the intracellular scaffolding proteins ZO-1, ZO-2 and ZO-3 [20]. The occludin is known to modulate blood–brain barrier integrity and functionality after stroke [18]. 

The opening of the BBB and subsequent infiltration of serum components to the brain can lead to a host of processes resulting in progressive synaptic neuronal dysfunction, and detrimental neuroinflammatory changes [8]. In OSA, repetitive hypoxic and reoxygenation episodes stimulate endothelial cells at the BBB, resulting in plasma proteins leaking into the arteriolar walls and perivascular spaces. Vasogenic edema [21] is also associated in the hippocampus area, proving that there is water transport from the blood compartment to the brain. Edema develops as a consequence of the breakdown of the BBB and is thought to be mediated by specific water channels in cell membranes, namely aquaporins (Aqps). Aqps are a family of water channels embedded in the plasma membrane of astrocytes and endothelial cells, and they are responsible for the flow of water in and out of the cells [16,17,22].

In the literature, it has been described that cerebellum cells are sensitive to intermittent hypoxia (IH)-induced lesions in a mouse model [23] characterized by a passage of water mediated by AQPs from the blood compartment to the interstitial area of the cerebral parenchyma and is indicative of a rupture of the BBB. These authors revealed that mice exposed to chronic IH for 35 days developed brain edema with altered Aqp1 and Aqp3 expression in the brains of an OSA model, indicating an important role of Aqps in chronic IH-mimicked OSA. In our previous published work, we assessed the time course of IH-induced neuroinflammation in animals exposed up to 24 weeks to mimic chronic sleep apnea/hypopnea in humans [24]. This type of edema is characterized by a passage of water, electrolytes and proteins from the blood compartment to the interstitial sector of the cerebral parenchyma and may be indicative of a rupture of the tight junctions of the BBB.

Based on the literature and on our previous results, we put forward the hypothesis that IH might play a major role in BBB alteration. We proposed to test this hypothesis in mice by studying the deleterious effects of IH on the BBB after 1 day and 15 days of exposure, in an experimental model of severe OSA. We focused on the dorsal part of the hippocampus, a highly vulnerable brain region implicated in impaired cognitive functions in OSA patients and a mouse model of OSA and we attempted to identify the underlying molecular mechanisms by combining BBB permeability tests (use of Evans blue marker). The passage of Evans blue from all the experiments was carried out on male mice at 10 weeks of the C57Bl6/J strain from the Janvier farm and carried out on the premises approved by the departmental veterinary bloodstream to the cerebral parenchyma induced by IH can occur paracellularly between endothelial cells via rupture of tight and adherent junctions, as well as by transcytosis. In this work, we studied the effects of IH on the molecular actors of the tight and adherens junctions of the BBB with RTqPCR assay. The genes that have been retained are those coding for the following: transmembrane proteins constituting the main elements of tight junctions: claudins 1, 5 and 12 (Cldn 1, Cldn 5, Cldn 12); occludin (Ocln); cytoplasmic proteins zona occludens 1, 2 and 3 (Zo1, Zo2, Zo3) that link Cldns to intracellular actin filaments via cingulin (Cgn); VE-Cadherin (Cdh5), the transmembrane protein constituting adherens junctions [25].

## 2. Results

In the first step, we measured the brain weight in our murine model after 15 d of IH exposure. The reference weight brain was 0.45 ± 0.01 g. We observed that IH mice groups had a similar brain weight of 0.43 ± 0.01 g than normoxic mice one of 0.44 ± 0.01 g. The body weight measurement was performed after 15 d with either intermittent hypoxia (IH) or normoxia (NO). After 15 days, the IH mice groups had a lower body weight 23.6 ± 0.3 g than normoxic ones of 26.3 ± 0.3 g (*p* < 0.05). IH exposure significantly induced a moderate 10% loss of body weight, an expected observation in such experimental conditions.

We quantified the extravasation of Evans blue in mice after IH (Figure 1). The Evans blue permeability test showed no change in the permeability of the BBB after 1 d of exposure to IH (*p* = 0.194). Indeed, the mean concentrations of Evans blue were 3.27 µg/mg of brain tissue in mice of group NO/1 d and 3.57 µg/mg of brain tissue in mice of group HI/1 d (Figure 1).

In contrast, the Evans blue permeability test revealed a significant increase in BBB permeability in mice exposed to 15 d of IH compared to control (NO) mice (*p* < 0.01) and to those exposed for 1 d of IH (*p* < 0.001) (Figure 1). We found 2.63 µg of Evans blue/mg of brain tissue in mice of the NO/15 d group compared to 5.06 µg/mg of brain tissue in mice of the IH/15 d group.

We also evaluated the effect of IH on mRNA expression of AQPs 1, 4 and 9 (Aqp1, Aqp4, Aqp9) (Figure 2). One-day IH exposure significantly increased Aqp1 gene expression in the dorsal hippocampus (NO/1 d = 1.00 ± 0.17 vs. IH/1 d = 1.64 ± 0.21; * *p* < 0.05) (Figure 2).

Fifteen days of exposure to IH significantly increased Aqp1 (NO/15 d = 1.07 ± 0.19 vs. IH/15 d = 1.56 ± 0.23; * *p* < 0.05) and Aqp9 (NO/15 d = 0.79 ± 0.19 vs. IH/15 d = 1.26 ± 0.23, * *p* < 0.05) (Figure 2). There was no detectable difference in the expression levels of Aqp 4 gene.

Claudins 1, 5 and 12 (Cldn 1, 5 and 12), occludin (Ocln) and VE-cadherin (Cdh5) mRNA expression were evaluated in the dorsal hippocampus of mice exposed to 1 d and 15 d of intermittent hypoxia compared to control mice exposed to normoxia.

One to 15 days of IH increases claudin, occludin and VE-cadherin gene expression (Figure 3). Acute IH seems to significantly impact Cldn1 mRNA expression within the dorsal hippocampus since it doubled with 1 d of IH and the increased level of expression persists after 15 d of IH but with a less severe overexpression (*p* < 0.001 IH/1 d vs. NO/1 d; NO/1 d = 1.00 ± 0.07; IH/1 d = 2.20 ± 0.37; NO/15 d = 1.09 ± 0.12 and IH/15 d = 1.35 ± 0.23) (Figure 3). We also observed an increase in Cldn12 mRNA levels in animals exposed to 1 d and 15 d of IH (*p* < 0.05 IH/1 d vs. NO/1 d; *p* < 0.05 IH/15 d vs. NO/15 d; NO/1 d = 1.00 ± 0.06; IH/1 d = 1.32 ± 0.04; NO/15 d = 1.07 ± 0.09 and IH/15 d = 1.35 ± 0.12) (Figure 3).

Among the other transmembrane proteins of the tight junctions of the BBB, occludin also had slightly increased mRNA expression with 1 d of IH (*p* < 0.05 IH/1 d vs. NO/1 d; NO/1 d = 1.00 ± 0.06; IH/1 d = 1.33 ± 0.09; NO/15 d = 1.05 ± 0.07 and IH/15 d = 1.24 ± 0.11) (Figure 3).

Finally, mRNA expression of VE-cadherin slightly increased after 1 day of IH but not after 15 days (*p* < 0.05 HI/1 d vs. NO/1 d; NO/1 d = 1.00 ± 0.04; HI/1 d = 1.28 ± 0.11; NO/15 d = 1.09 ± 0.08 and HI/15 d = 1.24 ± 0.05) (Figure 3). The gene expressions of ZO proteins, Claudin 5 and cingulin remained unchanged (Figure 3 and Figure 4).

## 3. Discussion

In this paper, we put forward the hypothesis that IH might play a major role in BBB alteration. We proposed to test this hypothesis in mice by studying the deleterious effects of IH on the BBB after 1 d and 15 d exposure. We have chosen the dorsal hippocampal tissue because this area is implicated in impaired cognitive functions and has been well studied in several model including sleep apnea and hypoxia. We attempted to identify the molecular mechanisms by combining in vivo BBB permeability tests (Evans blue tests) and mRNA expression of several tight junction proteins (ZO (1, 2, 3), VE-cadherin, claudins (1, 5, 12), cingulin and aquaporins (1, 4, 9)) on hippocampal brain tissues. We observed in vivo a passage of Evans blue from the bloodstream to the cerebral parenchyma induced by IH. This may result from a paracellularly passage between endothelial cells via a rupture of tight and adherens junctions. This was confirmed, we observed that endothelial microvascular brain aquaporin 1 and 9 is overexpressed after 15 days of HI with a rapid increase in expression for aquaporin 1 after the first exposition. We also observed an increased expression of claudin-1. These BBB alterations are possibly associated with a vasogenic edema that may affect brain functions and accelerate neurodegenerative processes as observed in the literature [23].

OSA is a major health problem characterized by repeated episodes of hypoxia–reoxygenation. IH is associated with cerebrovascular risk and cognitive disorders. In our previous work, we exposed mice to 1 day or 6 or 24 weeks of IH (alternating 21%–5% FiO_2_ every 30 s, 8 h/d) or normoxia [24]. A hypothesis has emerged in recent years that cognitive impairment may be linked to the opening of the BBB and leading to neuronal alteration associated with brain oedema. Thus, we decided to investigate in a 10-week-old male mouse model exposed to acute IH at 1 d and chronically for 15 d. We observed a loss of body weight and a brain weight decrease. These results tend to demonstrate a stress in the animals as reported in the literature [23].

Kim et al. [26] also worked on C57BL6 mice model exposed to intermittent or sham hypoxia, alternating 30 s of progressive hypoxia and 30 s of reoxygenation for 8 h/d. After 14 d, animals exposed to IH showed hypomyelination in cerebellum white matter and higher serum levels of endothelin-1. The short- and long-term memories with an object recognition test were impaired in the group exposed to IH as compared to controls. No differences were found in the BBB permeability between the groups contrary to what we observed using Evans blue marker.

In our work, we observed a significant increase in BBB permeability, evaluated with Evans blue in mice exposed to 15 d of IH compared to control mice exposed to normoxia and those exposed to 1 d of IH. Our results agree with what has been observed in the literature after a longer time of IH. Aquaporins are a family of water channel proteins that are involved in regulating transcellular or transepithelial water movement [17,27,28]. In our results, we observed an increase in Aqp9 and Aqp1 mRNAs after 2 weeks. In the brain, it has been considered that Aqp1 is expressed only in the choroid plexus but not in endothelial cells in normal human and animal brains. However, Aqp1 was expressed in microvessel endothelia in human astrocytoma and metastatic carcinomas [23]. Aqp1 was also detected in endothelial cells in glioblastoma transplanted into mouse brain [29]. In addition, Aqp1 immunoreactivity was present in the endothelium of a few microvessels in the normal human brain [29] and a tissue microarray study showed the presence of Aqp1 in the microvessels in the human cerebrum and cerebellum [30]. We cannot exclude that in rodents, low levels of Aqp1 were found in brain microvessels induced by biostress like hypoxia. Baronio et al. [23] revealed that mice exposed to chronic IH for 35 d appeared to have brain edema along with altered Aqp1 and Aqp3 expression in the hippocampus of an OSA murine model. These increases are in line with the rise in permeability observed with Evans blue.

Aqp4 is the most predominant aquaporin in the brain and mainly expressed at the endfeet of astrocytes in the CNS [27]. Recent evidence has shown that Aqp4 is not only involved in water homeostasis and edema formation [31,32], but it also contributes to neuroinflammation. Considering that chronic IH exposure also induces reactive gliosis and Aqp4 is mostly expressed on the astrocytes, we speculated that altered astrocytic Aqp4 expression may be involved in chronic IH-induced brain injury. No difference in Aqp4 was observed in our 15 d IH exposition model. We could not exclude that in a chronology of aquaporin expression, Aqp4 would be increased later.

It is well known that endothelial cells of cerebral blood capillaries are interconnected by tight junctions and adherens junctions which form one of the main elements of the BBB. In this work, we decided to measure the expression of the mRNAs of the main actors of these junctions in order to evaluate the impact of IH on the BBB integrity.

IH seems to significantly impact Cldn1 mRNA expression within the dorsal hippocampus since it is doubled with 1 d of IH and and also after 15 d of IH. This result is in line with those reported in the literature. Sladojevic et al., 2019 [15], reported a significant upregulation of Cldn1 mRNA and protein, a nonspecific claudin for blood vessels, and downregulation in Cldn5 expression in a stroke murine model. Rarely expressed at the normal BBB, Cldn1 is expressed in pathological conditions. They identified that Cldn1 is highly expressed in leaky brain microvessels. These results revealed that Cldn1 is incorporated in the BBB tight junction complex, impeding BBB recovery and causing BBB leakiness during poststroke recovery. The stress corresponding to exposure to intermittent hypoxia is much less intense and the role of this junction protein in protecting barrier function can be seen very differently than in the context of stroke. Our results are in line with these data and would probably correspond to a compensatory mechanism observed in blood vessels in the case of acute hypoxia. Pfeiffer et al. [27] had also observed the protective role of Cldn1 in an experimental knockout Cldn1 transgenic C57BL/6 murine model in the context of autoimmune encephalomyelitis (EAE). The mice expressing Cldn1 exhibited a reduced disease burden during the chronic phase of EAE compared to control littermates. We also observed an increase in claudin 12 (Cldn12) expression. Cldn12 mediates the interaction of claudins with the cytoskeleton, by allowing the binding to intracellular scaffolding proteins ZO-1, ZO-2 and ZO-3. However, its potential contribution to BBB tight junctions remains unknown. It has been suggested in the literature that Cldn12 expression is not required for maintaining BBB integrity [33]. However, lack of Cldn12 promoted broad neurological and behavioural changes. Cldn12 expression in the CNS is not upregulated under inflammatory stimuli and its absence does not impair BBB integrity. This suggests that the exact mechanisms underlying this phenotype need to be investigated. A slight increase in occludin mRNA was observed in acute hypoxia (1 day) condition in line with those reported in the literature. Occludin is a key structural component of the BBB that has recently become an important focus of research in BBB damage. Many studies have demonstrated that occludin could regulate the integrity and permeability of the BBB. In our previous study on an in vitro rodent model bend3, we found an upregulation of occludin after hypoxia [34]. The function of the BBB depends on the level of occludin protein expression in brain endothelial cells [34]. We could also make a parallel with our recent work on bend3 BBB in vitro model subjected to intermittent hypoxia cycles. We showed that an alteration in the endothelial characteristics of the BBB model is progressive along the repetition of the three stress hypoxia cycles [35]. In particular, we observed a gradual increase in apparent Na-Fl permeability, which was related to a progressive decrease in the content of tight junction proteins, such as ZO-1 and claudin-5. This opening implies the possible passage of potentially toxic compounds from blood to brain and vice versa.

We also observed an upregulation in P-gp protein expression and functionality and a downregulation in BCRP [35,36,37]. We could not exclude that an adaptative mechanism exists to avoid BBB breakdown linked to other mechanisms involving other proteins like ABC channel membrane transporters, as observed in our in vitro studies [35]. We also observed an adaptative mechanism in our previous study upon effect of sleep apnea serum patients on an in vitro human BBB opening model [38]. ABC transporter may constitute an early protection response to injury in endothelial cells [36].

Contrary to what we expected, short term IH of 15 d led to an overexpression of mRNAs of molecular actors of tight junctions and adherens junctions of the BBB. In the current state of our experiments, we can hypothesize an adaptive phenomenon intended to compensate for a deleterious effect of IH on an element of the BBB not explored by our study for the moment.

Our study has limitations. We assessed Aqps and protein junction alterations through their mRNA expression. However, it is well known that the correlation between mRNA and protein levels may be modest. Therefore, mRNA results need to be confirmed by the evaluation of the corresponding proteins. We focused the study on the hippocampus as this brain region is involved in cognition and is susceptible to IH. We did not assess behavior because it was beyond the scope of the study, but it would be interesting to parallel behavioral findings and kinetics of cytokine mRNA and microglial alterations. The results of Kim et al. [26] indicated that hypomyelination and impairment of short- and long-term working memories occurred in C57BL mice after 14 d of IH-mimicking sleep apnea, the same way we could observe in our protocol. All these alterations may affect cognitive functions and aggravate neurodegenerative processes. Therefore, in light of these findings and in addition to other IH-related consequences such as vascular, neuronal, astroglial and sleep alterations, IH contributes to pathological brain aging and is essential to detect and treat as early as possible.

## 4. Materials and Methods

### 4.1. Murine Model of IH

The protocol was approved by the institutional committee of Grenoble for animal care and use (Protocol number: 56_UHTA-HP2-MD-01). Stalling and exposure to IH were carried out in rooms lit according to a day/night cycle of 12 h/12 h (light phase from 07:00 to 19:00) and the temperature was regulated (22 ± 2 °C). Animals were provided with water and standard rodent chow ad libitum. On their arrival at the animal facility (Platform of High Animal Technology, Jean Roger building at the Faculty of Medicine and Pharmacy of Grenoble), the mice, aged 8 weeks, were randomly placed into groups in cages placed on a ventilated rack (6 mice per cage).

After 1 week of housing in the animal facility, the cages were placed in the IH device. This was obtained by intermittent enrichment of the air with nitrogen up to a fraction inspired in 5% oxygen (FiO_2_), followed by up to 21% reoxygenation every minute. Using this device developed in the HP2 laboratory (Inserm U1300, Grenoble) [24], the animals remained in groups in their cages during the IH exposure. This avoided stress related to isolation and change in environment. In parallel, control animals were placed on the same rack, but exposed to normoxia (NO) (constant 21% FiO_2_) ensuring similar turbulence and noise between IH and NO conditions. IH or N stimulus was automatically applied 8 h/d (from 07:00 to 15:00, which corresponds to the phase during which rodents sleep preferentially) for 15 days. FiO2 variations were checked daily using a gas analyzer. In order to ensure the good health of the animals, the mice were weighed on D-1, D0, D1, D6 and the day of sacrifice.

The animals were anesthetized with sodium pentobarbital immediately after being removed from the IH system. Fresh brain tissues were weighed and placed in liquid nitrogen for real-time semiquantitative polymerase chain reaction (RT-PCR) analysis and stored at −80 °C.

### 4.2. Evaluation of BBB Permeability In Vivo

A total of 48 mice were included in the BBB permeability analysis (12 IH mice/1 d, 12 NO/1 d, 12 IH/15 d and 12 NO/15 d) (Figure 5). One day before sacrifice, the mice received an intraperitoneal injection of a 2% Evans blue solution (4 mL/kg; Sigma Aldrich, Saint Louis, MO, USA). During sacrifice, the animals were deeply anesthetized and the brains were removed on ice, weighed and then ground on ice in distilled water using a polytron. A protein assay was carried out on the ground material. The rest of the ground material was centrifuged and absorbance was read on the supernatant with a spectrophotometer at 610 nm. A standard range was produced with known concentrations of Evans blue diluted in this supernatant. The results were obtained (μg of Evans blue/mg of brain tissue) and expressed as mean ± SEM (standard error of the mean or standard error) and then analyzed using a two-way ANOVA (SigmaPlot 12.0) where the significance threshold was set at 0.05.

### 4.3. Relative Measurement of mRNA Expression of BBB Actors by Quantitative RT-PCR

A total of 24 mice were included in the PCR analysis of BBB molecular markers (6 IH mice/1 d, 6 NO/1 d, 6 IH/15 d and 6 NO/15 d): aquaporin 1, 4, 9, claudin 1, 5, 12, occludin, cingulin, zonula occludens 1, 2, 3 and VE-cadherin 5. After deep anesthesia of the animals, the brains were quickly removed on ice and immediately placed at −80 °C until tissue processing. The dorsal hippocampus was removed under binocular magnifying glass using a scalpel and then stored at −80 °C until the extraction of total RNA. The complementary DNAs (cDNAs) were obtained by reverse transcription from the total RNAs according to our previous method described in our paper [24]. The results were processed using CFX Manager analysis software 2.1 (Biorad Laboratories, Hercules, CA, USA). Beforehand, the effectiveness of PCR was tested for each pair of primers (Table 1) using a cDNA standard range (cDNA diluted to 1/10, 1/20, 40, 1/80 and 1/160). The efficiency must be close to 100% (acceptable between 95–105%). The level of expression of the genes was analyzed according to the comparative method of the cycle threshold (Ct). The results were expressed using the relative quantification method of 2^−∆∆Ct^. Here we made a correction from 5 reference genes (Rplp0: ribosomal protein large P0; Rpl13A: ribosomal protein L13A; Hprt1: hypoxanthine phosphoribosyltransferase 1; Ywhaz: tyrosine 3-monooxygenase/tryptophan 5-monooxygenase activation protein, zeta polypeptide; Actb: beta actin) for which the levels of mRNA expression do not vary according to the experimental conditions. The mean mRNA expression of a gene of interest for the NO/1 d animals was set at 1. The results of relative mRNA expressions were expressed as mean ± SEM and analyzed using a two-way ANOVA.

### 4.4. Statistical Analysis

Statistical analysis was performed using SigmaPlot software and a two-way ANOVA or Student’s t-test, where the significance level was set at 0.05. The differences between means were considered significant when *p*-values were <0.05. The values of the variables are expressed as means ± SEM.

## 5. Conclusions

These results revealed that AQPs that regulate water passage at the origin of oedema in the brain were increased after IH exposure, especially Aqp1 and 9 but not Aqp4. Cldn1 and 12, VE-cadherin are implicated in the immediate response to IH. To put in light a potential compensatory mechanism, protein expressions have to be evaluated and completed with ABC transporter expressions that prevent xenobiotics entering through the endothelial cells.

## Figures and Tables

**Figure 1 ijms-25-03065-f001:**
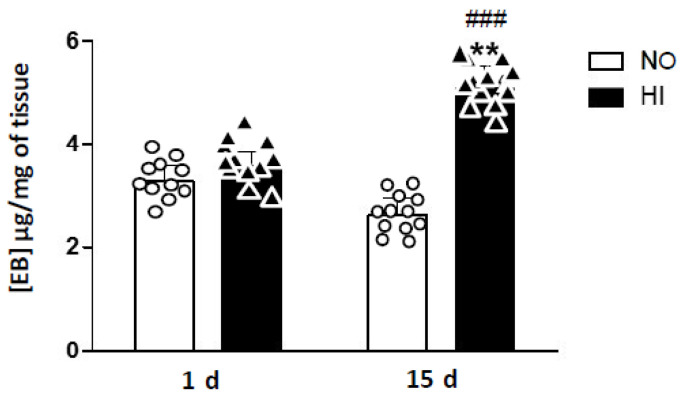
Blue Evans experiments. Increase in blood-brain barrier permeability in mice exposed to 15 days of intermittent hypoxia (IH) compared to control mice exposed to normoxia (NO) and to those exposed to 1 day of IH. The results are expressed as the concentration of Evans blue in the ground brain tissue ([EB] µg/mg of tissue). n = 12 in each group, Student *t*-test ** *p* < 0.01 IH/15 d vs. NO/15 d; *^###^ p* < 0.001 IH/15d vs. IH/1 d.

**Figure 2 ijms-25-03065-f002:**
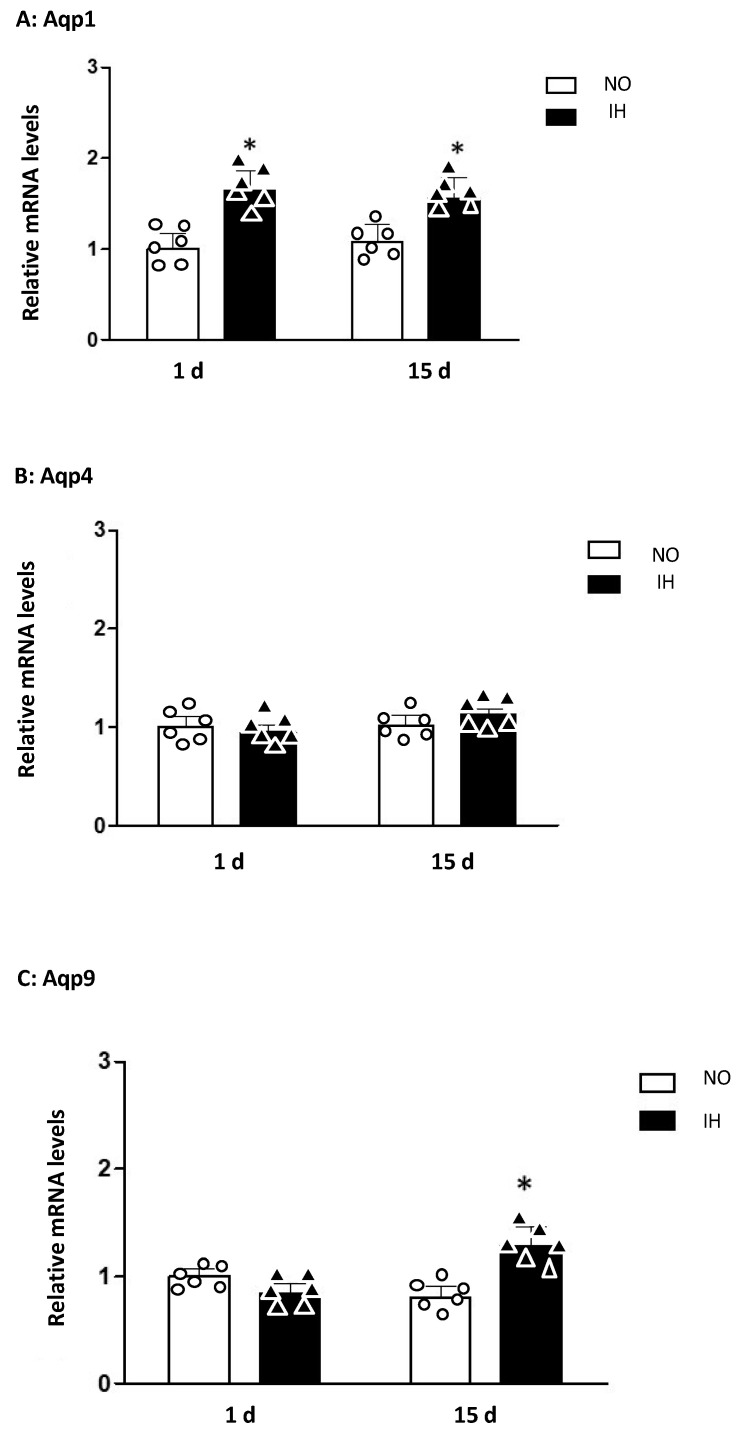
Aquaporins. Expression of mRNAs of aquaporins (Aqp) 1 (**A**), 4 (**B**) and 9 (**C**) in the dorsal hippocampus of mice exposed to 1 d and 15 d of intermittent hypoxia (IH), compared to control mice exposed to normoxia (NO). The results are expressed in arbitrary units (a.u.). n = 6 in each group, mean ± SEM (* *p* < 0.05 IH vs. NO). All values were normalized to Normoxia 1 d.

**Figure 3 ijms-25-03065-f003:**
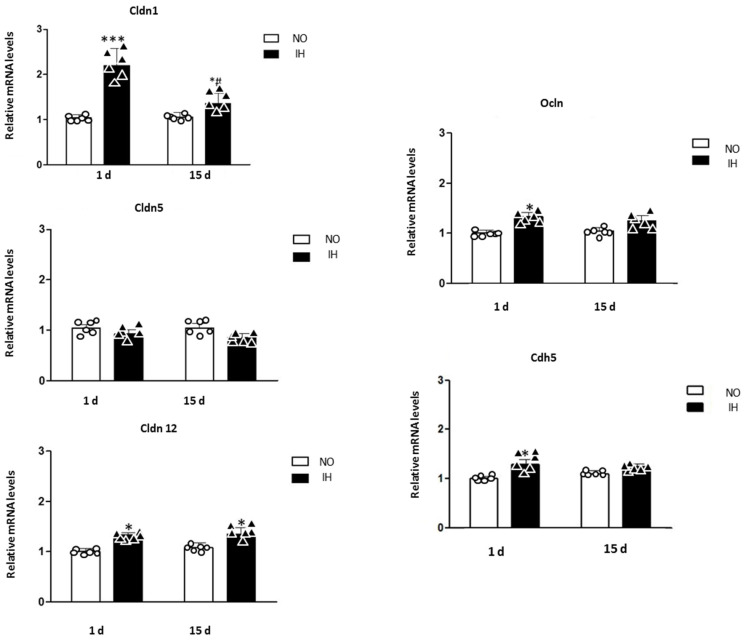
Claudin Occludin VE cadh. Claudins 1, 5 and 12 (Cldn1, Cldn5 and Cldn12), occludin (Ocln) and VE-cadherin (Cdh5) mRNA expression in the dorsal hippocampus of mice exposed to 1 d and 15 d of intermittent hypoxia (IH) compared to control mice exposed to normoxia (NO). The results are expressed in arbitrary units (a.u.). n = 6 in each group, mean ± SEM * *p* < 0.05 IH/15 d vs. NO/15 d; *** *p* < 0.001 IH/15 d vs. NO/15 d; *^#^ p* < 0.05 IH/15 d vs. IH/1 d.

**Figure 4 ijms-25-03065-f004:**
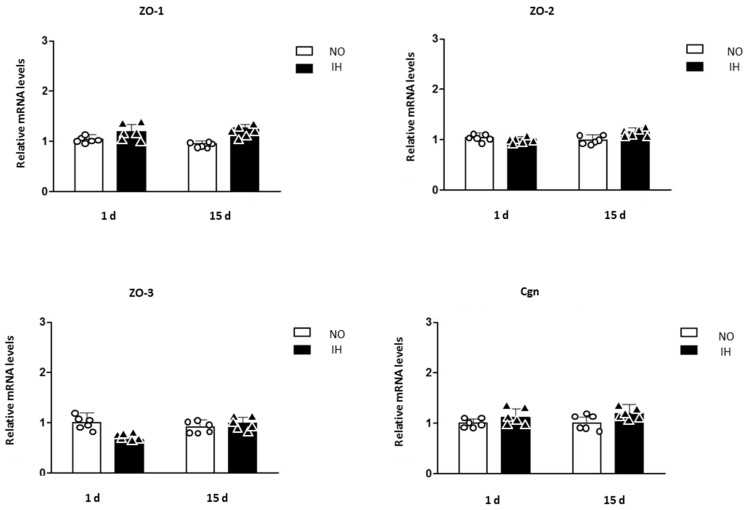
ZO. Zonula Occludens (ZO-1, ZO-2, ZO-3) and Cingulin mRNA expressions in the dorsal hippocampus of mice exposed to 1 d and 15 d of intermittent hypoxia (IH) compared to control mice exposed to normoxia (NO). The results are expressed in arbitrary units (a.u.). n = 6 in each group, mean ± SEM.

**Figure 5 ijms-25-03065-f005:**
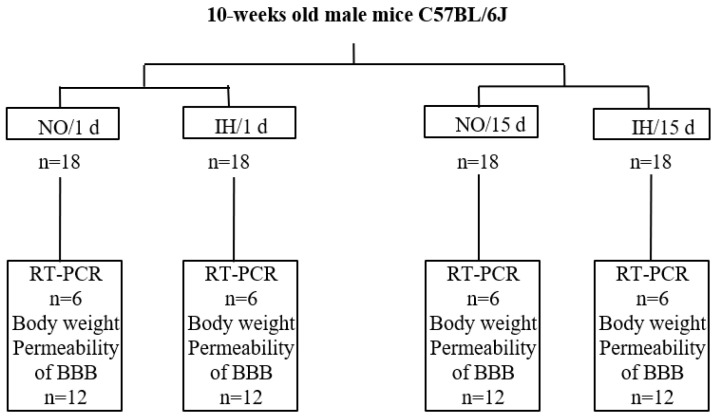
Flowchart of the experiments. (NO: normoxia, IH: intermittent hypoxia, BBB: blood brain, barrier).

**Table 1 ijms-25-03065-t001:** Sequence of primers used in the study.

Gene		Sequence 5′-3′
*Aqp 1*	Forward	CTGCTGGCGATTGACTACACTGG
	Reverse	CCCCACCCAGAAAATCCAGTG
*Aqp 4*	Forward	GGTGGGGTAAGTGTGGACATTCC
	Reverse	AGCGTGGCCAGAAATTCTGCT
*Aqp 9*	Forward	CATCACGGGAGAAAATGGAACG
	Reverse	GAGGAACATGGTAGACACCACTTGG
*Cldn 1*	Forward	GTTTGCAGAGACCCCATCAC
	Reverse	AGAAGCCAGGATGAAACCCA
*Cldn 5*	Forward	TCTGCTGGTTCGCCAACAT
	Reverse	CGGCACCGTCGGATCA
*Cldn 12*	Forward	ATCACATTCAACAGAAACGAG
	Reverse	GGTCGTACATCAGGCAGTCA
*ZO-1*	Forward	TTAAGGACAAATTCCGCAGGCTAC
	Reverse	TCTCAGCCTTCTTGATCAGTGCG
*ZO-2*	Forward	GATAGCAGCCATCGTGGTCAAGA
	Reverse	GCGGAAACTTCTGCCATCAAATT
*ZO-3*	Forward	CCAGTTTTAAGCGCCCCGTG
	Reverse	AGTCCGGGACATGCTTTCTGC
*Cingulin*	Forward	GCAGTCCACCAATCGCAAACTAG
	Reverse	GCCTTCACCCTCAGGGTTAGCT
*Cadherin 5*	Forward	GTACAGCATCATGCAGGGCGA
	Reverse	CTTCATAGTCCAGGGACTTCGTGG

## Data Availability

Data are contained within the article.

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
