# Peer review of "Early Increase in Blood–Brain Barrier Permeability in a Murine Model Exposed to Fifteen Days of Intermittent Hypoxia"

_ijms, 2024, doi:10.3390/ijms25053065_

Round 1
Reviewer 1 Report
Comments and Suggestions for Authors
The authors present an interesting study involving
Several major concerns that need attention
1. Authors provide abbreviations without adequate description. For example ZO-zonula occludens
2. Hypothesis provided without adequate explanation
3. Please elaborate: ''The opening of the BBB and subsequent infiltration of serum components to the brain can lead to a host of processes resulting in progressive synaptic neuronal dysfunction, and detrimental neuroinflammatory changes''
4. Provide reference: In OSA, repetitive hypoxic and reoxygenation episodes stimulate endothelial cells at the BBB, resulting in plasma proteins leaking 73 into the arteriolar walls and perivascular spaces
5. Explain other effects of intermittent hypoxia due to OSA
6. Justification of the study and future recommendations
Comments on the Quality of English LanguageThe sentence is difficult to read and appears confusing.
Author Response
Dear Editor,
>
>
> Please Find enclosed the corrected version as you requested in a file joined with figures and graphical abstract.
>
> •We have entirely revise text formatting. The text, legends, and tables use several fonts and dimensions has been checked. We remove underscores, uniform the color of the characters and adjust the left border.
> • The list of all the abbreviations used have been verified and in alphabetical order.
> • we Uniform paragraph headers as they are sometimes in upper case and other times in normal cases.
>
> We have checked and remake all the figures, we have put them in a separate file note in text in PDF and PPT form to correct all the problems raised i.e :
> • Figure 1 and split into on page.
> • Bar graphs must report all the individual data points.
> • Use dots, to identify the decimal point.
> • the number of digits on the right of the decimal point.
> • We modify the graphical abstract
> I hope that all the corrections will be satisfactory
> Sincerely yours

Reviewer 2 Report
Comments and Suggestions for Authors
The manuscript entitled Early increase in blood-brain barrier permeability in a murine model exposed to fifteen days of intermittent hypoxia (IH) is an original article. The authors studied possible consequences on the blood-brain barrier (BBB) in patients with OSA. OSA is a public health problem and CPAP is the most efficient method of treatment. It is important to study the intimate mechanisms of the IH during OSA. The background and the aims of the article are finely defined.
The methodology is rigorous and clearly explained. The statistical analysis is adequate.
The results are quite interesting. Two weeks of IH decreases the brain weight and the body weight of the induced a 10% loss of the body weight. The authors find that Cldn1 is the immediate response to IH. AQPs that regulate water passage through BBB were increased, especially Aqp1 and 9 but not Aqp4.
Some observations to improve the quality of the manuscript
Page 1: The authors have to insert the abbreviation for intermittent hypoxia from the first mention in the text, including the abstract.
Page 2: The authors have to insert the abbreviation for tight junctions (TJs) from the first mention in the text.
Page 3: We suggest editing the text for’’ The genes that have been retained are those coding for the following:’’ in a continuous format.
Page 6: The words ‘’Evans blue’’ should be written with regular characters.
In the section on Discussions, the authors should mention some thresholds of severity for their found markers in IH from OSA. Because cephalometry is very important for the hypoxia index in OSA. This particular mention in the manuscript could adjust the quality of discussions in the manuscript
Further research is necessary to study transporter protein expressions in BBB in connection with IH from OSA. Also, the authors can develop in further studies the effects of the short-term IH which led in the present study to an overexpression of mRNAs of proteins from tight junctions and adherens junctions of the BBB.
Author Response
Dear Editor,
>
>
> Please Find enclosed the corrected version as you requested in a file joined with figures and graphical abstract.
>
> •We have entirely revise text formatting. The text, legends, and tables use several fonts and dimensions has been checked. We remove underscores, uniform the color of the characters and adjust the left border.
> • The list of all the abbreviations used have been verified and in alphabetical order.
> • we Uniform paragraph headers as they are sometimes in upper case and other times in normal cases.
>
> We have checked and remake all the figures, we have put them in a separate file note in text in PDF and PPT form to correct all the problems raised i.e :
> • Figure 1 and split into on page.
> • Bar graphs must report all the individual data points.
> • Use dots, to identify the decimal point.
> • the number of digits on the right of the decimal point.
> • We modify the graphical abstract
> I hope that all the corrections will be satisfactory
The text has been uploaded in a the response to reviewer 1 and the figures are uploaded in the response to reviewever 2
> Sincerely yours

Round 2
Reviewer 1 Report
Comments and Suggestions for Authors
The revisions has been made adequately